# A herbicide resistance risk assessment for weeds in wheat and barley crops in New Zealand

Zachary Ngow[1]☯*, Richard J. Chynoweth[2‡], Matilda Gunnarsson[2‡], Phil Rolston[2‡], Christopher E. Buddenhagen[1]☯*

**1** AgResearch Ltd., Hamilton, New Zealand, **2** Foundation for Arable Research, Christchurch, New Zealand

☯ These authors contributed equally to this work.
‡ These authors also contributed equally to this work.
* zacharyngow@gmail.com (ZN); chris.buddenhagen@agresearch.co.nz (CEB)

**Data Availability Statement:** All relevant data are within the paper and its Supporting Information files.

## Abstract

We estimated the risk of selecting for herbicide resistance in 101 weed species known to occur in wheat and barley crops on farms in New Zealand. A protocol was used that accounts for both the risk that different herbicides will select for resistance and each weed's propensity to develop herbicide resistance based on the number of cases worldwide. To provide context we documented current herbicide use patterns. Most weeds (55) were low-risk, 30 were medium-risk and 16 high-risk. The top ten scored weeds were *Echinochloa crus-galli*, *Poa annua*, *Lolium multiflorum*, *Erigeron sumatrensis*, *Raphanus raphanistrum*, *Lolium perenne*, *Erigeron bonariensis*, *Avena fatua*, *Avena sterilis* and *Digitaria sanguinalis*. Seven out of ten high-risk weeds were grasses. The most used herbicides were synthetic auxins, an enolpyruvylshikimate-phosphate synthase inhibitor, acetolactate synthase (ALS) inhibitors, carotenoid biosynthesis inhibitors, and long-chain fatty acid inhibitors. ALS-inhibitors were assessed as posing the greatest risk for more species than other modes-of-action. Despite pre-emergence herbicides being known to delay resistance, New Zealand farmers only applied flufenacet and terbuthlazine with high frequency. Based on our analysis, surveys for herbicide-resistant species should focus on the high-risk species we identified. Farmer extension efforts in New Zealand should address resistance evolution in cropping weeds.

## Introduction

A worldwide analysis suggests that weeds have the highest potential to cause yield losses in major crops including wheat and barley where they account for potential losses of 23% [1,2]. In response to this threat to production, farmers have come to rely on a mix of cultural and chemical control methods. Herbicidal weed control is favoured as it is cost-effective providing a three or four-fold economic return [2] and is a key element in the implementation of conservation tillage systems (including direct drilling) which reduce soil erosion [3]. However, intensive herbicide use is known to select for resistant individuals in weed populations, eroding herbicide effectiveness as these weeds escape control [4–6]. Globally farmers are addressing

**Funding:** All the authors worked under the Ministry of Business, Innovation and Employment [grant number C10X1806] to AgResearch Ltd.: "Improved weed control and vegetation management to minimize future herbicide resistance." AgResearch Ltd is a crown owned research institution doing science research businesses but owned by the Crown (i.e. the Government) in New Zealand. The funding agency is also the main public science funding organization in New Zealand and provided financial support in the form of authors' [ZN and CEB] salaries and/or research materials. They did not play a role in the study design, data collection and analysis, decision to publish, or preparation of the manuscript.

**Competing interests:** AgResearch Ltd is a crown owned research institution doing science research businesses but owned by the Crown (i.e. the Government) in New Zealand. ZN and CEB's affiliation to AgResearch Ltd. does not alter our adherence to PLOS ONE policies on sharing data and materials. The funding agency is also the main public science funding organization in New Zealand and provided financial support in the form of authors' salaries and/or research materials.

similar suites of weed species in any given crop, and some appear to have a greater propensity to develop resistance, with repeated convergent evolution of resistance being documented in the International Survey of Herbicide Resistant Weeds [4] and the scientific literature generally [5,6]. Over the last 40 years a total of 14 species of herbicide resistant weeds have been documented in New Zealand, with 12 instances of resistance documented in arable crops including maize, peas, oats, wheat, and barley [7]. In any given year >50% of New Zealand arable production areas are under wheat (~45000 ha) and barley (~55000 ha) rotations [8]. Other crops that may commonly be included with wheat and barley rotations are; pasture, spring-sown peas, linseed, ryegrass, clover, and oilseed rape. Production levels are high, with farmers in New Zealand obtaining world record wheat and barley yields in 2017 and 2015, respectively [9,10].

An examination of weed science publications shows that our knowledge about herbicide resistance cases has developed mostly via unsystematic detection globally. It reflects the varying effort, scientific input, and methods of detection; definitely this is the case in New Zealand [7,11]. This makes sense since farmers and herbicide companies do not necessarily report resistance cases to scientists, and even if they do, not all cases are likely to get published. What's more, ad-hoc reporting may reflect strong biases towards a small number of the most problematic weeds. With some notable exceptions, particularly in cropping systems in Australia [12–17], systematic surveys to detect herbicide resistance cases are rare, and often focus on one or two problematic species in a given crop, for example, *Alopecurus myosuroides* Huds. in French wheat fields [18] or *Avena fatua* L. in two Canadian townships [19]. Systematic surveys may be rare because of the cost, a lack of specific pathways for reporting [11], and industry perceptions about the importance of resistance. Surveyors should ideally be open to the discovery of completely novel cases while also being aware of those that are most at risk of developing resistance.

Here we adapt a recently published risk assessment protocol [20] to identify those weeds most at risk of developing resistance in New Zealand, given their occurrence in wheat and barley fields, and their prior record of resistance in wheat and barley fields elsewhere in the world. This Moss et al. protocol [20] (henceforth the "Moss protocol") set out to assess resistance risk as part of a pesticide authorization process in Europe, based on a European Plant Protection Organization (EPPO) protocol, originally developed in 1999 [21]. They present a quantitative risk matrix using both herbicide-risk (some herbicides pose a higher risk than others) and species-risk (some weed species are more resistance-prone than others), with an optional score modifier designed to account for agronomic management practices that may reduce the risk. We took advantage of a unique data set about herbicide use in wheat and barley fields in New Zealand to place our risk assessment into context, and construct a framework for herbicide resistance surveys and extension efforts in the New Zealand cropping industry. This risk assessment is on an industry-wide scale informed by anonymized herbicide application data from wheat and barley fields. Risks were not assessed at the scale of individual farms and fields, this requires detailed information about herbicide timing, mixtures and rotations, and their interactions with weed biology, crop rotations and other cultural practices. All the high-risk weeds identified here should be targeted in surveys designed to detect herbicide-resistant weeds.

## Materials and methods

### Weed list

We generated a list of potential target weeds from wheat and barley crops in New Zealand. This was primarily sourced from the Bourdôt et al. 1998 weed survey in New Zealand

(Canterbury region) wheat and barley fields [22]. The Canterbury and northern Otago regions contain more than 75% of all the wheat and barley grown in New Zealand [8]. We expanded the weed species list to include species known to occur in wheat and barley fields in the wider New Zealand context. Grasses and some broadleaf genera were not identified to species by Bourdôt et al. [22], hence we took steps to address this gap and other omissions by using other literature [23,24], expert knowledge and field observations made in January (late summer) of 2019 and 2020. Species nomenclature follows the New Zealand Flora and taxonomic authorities are listed in S1 Table. [25]. Subspecies were not distinguished, and taxa were considered by us only at the species level (e.g. *Avena sterilis* subsp. *ludoviciana* (Durieu) M.Gillet & Magne was treated as *Avena sterilis L.*).

## Ranking herbicide groups by resistance cases

We ranked Herbicide Resistance Action Committee (HRAC) legacy herbicide mode of action (MoA) groups by the number of resistance cases documented by the International Survey of Herbicide Resistant Weeds [4] to obtain an estimate of what Moss et al. called the "inherent risk" of the herbicide [20]. The Moss protocol [20] considers the inherent risk to relate to the total number of cases of resistance reported in the International Survey of Herbicide Resistant Weeds [4] for each legacy HRAC MoA [26], with risk scores of 1, 2 or 3 given for low, medium or high risk respectively. We set the threshold for high-risk at >9% of recorded cases, which captures the original high-risk categories identified in the Moss protocol, but now places group G (glyphosate) in the high category. Moss et al. used the 10% threshold for high-risk herbicides. With group A having 48 cases and group G herbicides having 47 cases we chose to place the two groups in the same risk category, with such a small difference in the numbers of cases worldwide we believe they are indistinguishable from the data. The alternative is to use the same threshold as in the Moss protocol, but this would result in group A and G being medium risk, which fails to capture the high-risk status of group A herbicides. There are different ways we can reasonably set risk thresholds, which will be discussed later. Remaining ranking thresholds were not changed: medium risk 5–9%, low risk 1–5% and very low risk <1%. Low-risk and very low risk are both scored as '1'.

## Herbicide use

The most recent (2017–2018) data on herbicide usage trends in New Zealand were sourced from the ProductionWise® [27] platform and aggregated by active ingredient. This consisted of data entered by farmers about herbicide use in 5026 barley and 7647 wheat fields. Approximately 900 arable farmers have registered to use the platform, but anonymization was complete, with no unique identifiers for farms or fields. Farmers recorded every spray event (by herbicide product) in their fields. For example, an individual active ingredient used three times in a field is recorded three times. Products with multiple active ingredients were recorded as independent applications. Counts of herbicide use in fields were summarized by active ingredient and legacy HRAC [26] herbicide mode-of-action, from product label information. Relative rates of use by mode-of-action were quantified and characterized as very high (>20% of all application instances), high (>10%), moderate (>1%), low (~1%), extremely low (<1%) and nonexistent (0%).

## Cases of resistance by taxon and risk scoring

We assume that the best way to predict resistance in a weed species to any given herbicide (by HRAC group) is proportional to the number of documented cases of herbicide resistance in the same taxon, given the use of the same herbicide type in New Zealand. To calculate Moss's

"inherent" species-risk scores we used the global number of resistance cases from the International Survey of Herbicide Resistant Weeds [4]. Cases are defined by the International Survey of Herbicide Resistant Weeds as unique combinations of weed species and HRAC herbicide mode-of-action (species x site of action). To obtain the 'high', 'medium' and 'low' risk scores as used in the Moss protocol [20], we designated ≥10 cases as high risk (score = 3), < 10 as moderate (score = 2) and no cases recorded as low risk (score = 1). We assessed overall species-risk as the sum of the herbicide-risk multiplied by the "inherent" species-risk [20] combined for all relevant HRAC MoA herbicide groups, but only, but only where species had cases of resistance documented somewhere in the world. We include all cases of resistance for each weed species, rather than restricting our focus to cases from wheat and barley, because we are interested in a species propensity to develop resistance to a herbicide group. For example, the high-risk species *Chenopodium album* L. has more than 10 documented cases of resistance giving it a species score of 3. Then we consider the herbicide-risk scores for those herbicides where *Chenopodium album* has evolved resistance somewhere in the world. There were cases in two high-risk herbicide groups B and C1 (each with a herbicide score of 3), and one medium-risk group O (herbicide score of 2). The species and herbicide scores are multiplied and summed (3×3) + (3×3) + (3×2) = 24. We distinguished cases that were in herbicide groups highly-used (or not) by wheat and barley farmers in New Zealand.

The summed (cumulative) scoring method described above is not used in the European Moss protocol because its purpose was to regulate herbicide product use [20]. In contrast, we wanted to determine the risk that different herbicides will select for resistance in weed species known to occur in New Zealand's wheat and barley fields. Ultimately we hope to inform sector stakeholders about risk, and to improve herbicide resistance detection. We adapted their protocol to score species-risk in an industry-wide assessment. Unlike Moss et al. [20] we used an explicit threshold (though arbitrary) to determine species-risk scores based on the number of cases of resistance worldwide, but we think this approach produces credible risk estimates in the light of current knowledge. We examined 101 species and ended up with 46 'high' and 'medium' resistance risk species, many more than Moss et al (they scored an example list of only 13 high and medium risk taxa). The Moss protocol also used score modifiers that take into account resistance management practices including the use of non-chemical control measures. We did not use the score modifiers since these vary from field to field and our objective was slightly different. We acknowledge that actual risk of resistance development is determined mostly by the frequency and type of herbicide applied (selection pressure) interacting with characteristics of weed biology, distribution and abundance [5]. All graphs were created in R using the ggplot2 package [28,29].

## Results

### Weed list

Our weed list contained 69 species from the original Bourdôt et al. article [22]. An additional 32 species were added based on field observation, expert opinion and relevant literature [23,24], resulting in a total of 101 weed species for consideration. *Digitaria sanguinalis*, *Echinochloa crus-galli*, *Erigeron* spp. and *Raphanus raphanistrum* are notable emerging weeds, so they were included. Some taxa noted by Bourdôt were resolved to species, such as *Trifolium* spp., which became *Trifolium repens* and *Trifolium pratense*. A full list of weed species that we considered for New Zealand wheat and barley crops is displayed with taxonomic authorities in S1 Table.

## Ranking herbicide groups by resistance cases

Herbicide mode-of-action risk rankings are displayed (Table 1), arranged from high risk to very low risk. This resembles the table of Moss et al., [20] with a notable exception in that group G is raised to high risk. We designated HRAC groups B, C1, A, G as high-risk (>9% of recorded cases), O, D, C2 as moderate (5–9%), E, K1, K3, N F, Z as low (1–5%) and C3, F1, H, L as very low risk (<1%).

## Herbicide use

Current herbicide use in New Zealand wheat and barley fields involves 75 unique active ingredients in 16 mode-of-action groups (we show the 11 most important herbicide groups in Table 2). Barley and wheat have shared patterns of herbicide usage with respect to mode-of-action (Table 2) and active-ingredients. Synthetic auxins (HRAC group O) were represented in higher proportions than any other class of herbicide (a total usage rate of 26%). ALS-inhibitors (B), PDS-inhibitors (F1) and EPSPS-inhibitors (G) were highly used herbicide groups with >10% total usage, and acetyl coenzyme-A carboxylase inhibitors (A) and photosystem-II disrupters (C) were moderately used (total <10%). EPSPS-inhibitors (G) were used in larger proportions in barley (18%) compared to wheat (12%); conversely, farmers used K3 herbicides significantly more in wheat at 12% compared to barley at 4%. Glyphosate (MoA group G) is mostly used (>95%) used to control weeds pre-sowing of the cereal crops, for termination of the previous crop or pre-establishment weed control. It is very rarely used as crop pre-harvest desiccant. The following 10 HRAC MoA groups each accounted for less than 1% of herbicide applications, N, F4, K1, I, H, F3, Z, L, F2 & K2.

**Table 1. Herbicide mode-of-action groups ranked by resistance risk.** The number of cases of resistance from the International Survey of Herbicide Resistant Weeds [4] ranked and grouped by the legacy HRAC mode-of-action (data accessed in January 2020).

| Resistance Risk | | HRAC Herbicide MoA Groups | Example active ingredient | Number of resistant species worldwide | % of the worldwide total |
|---|---|---|---|---|---|
| High | B | ALS inhibitor | flumetsulam | 165 | 32 |
| | C1 | PSII inhibitors (triazines) | atrazine | 74 | 15 |
| | A | ACCase inhibitors | pinoxaden | 48 | 9 |
| | G | EPSP synthase inhibitors | glyphosate | 47 | 9 |
| Medium | O | Synthetic auxin | MCPA | 41 | 8 |
| | D | PSI electron diverters | paraquat | 32 | 6 |
| | C2 | PSII inhibitors (ureas and amides) | isoproturon | 29 | 6 |
| Low | E | PPO inhibitors | carfentrazone | 13 | 3 |
| | K1 | Microtubule inhibitors | trifluralin | 12 | 2 |
| | N | Lipid inhibitors | triallate | 10 | 2 |
| | K3 | Long-chain fatty acid inhibitors | flufenacet | 7 | 1 |
| | F3 | Carotenoid biosynthesis (unknown target) | amitrole | 6 | 1 |
| | Z* | Anti-microtubule mitotic disrupter | flamprop | 6 | 1 |
| Very-low | C3 | PSII inhibitors (nitriles) | ioxynil | 4 | <1 |
| | F1 | Carotenoid biosynthesis inhibitors | diflufenican | 4 | <1 |
| | H | Glutamine synthase inhibitors | glufosinate | 4 | <1 |
| | L | Cellulose inhibitors | dichlobenil | 3 | <1 |
| | † | Other MoA | - | 6 | 1 |

*Z includes subcategories Z1, Z2, Z3, Z4.

†Other includes mode-of-actions with 2 or fewer cases: F2, F4, K2, I.

**Table 2. Ranked herbicide mode-of-action usage proportions.** The percentage of fields that received herbicide applications, grouped by HRAC MoA categories (data are sourced from ProductionWise®, 2017–2018).

| Mode-of-action | Barley % | Wheat % | Total % |
|:---:|:---:|:---:|:---:|
| O | 30 | 24 | 26 |
| B | 16 | 17 | 16 |
| F1 | 14 | 17 | 16 |
| G | 18 | 12 | 14 |
| K3 | 4 | 12 | 9 |
| A | 8 | 6 | 7 |
| C1 | 3 | 5 | 4 |
| C2 | 1 | 4 | 3 |
| C3 | 2 | 1 | 1 |
| E | 2 | <1 | 1 |
| D | 1 | 1 | 1 |

Individual herbicide usage ranking for each crop is displayed in Fig 1, ranked by the wheat herbicide use. The ten most used herbicides (in order) for barley were: glyphosate (HRAC group G), diflufenican (F1), fluroxypyr (O), MCPA (O), chlorsulfuron (B), pinoxaden (A), iodosulfuron (B), flufenacet (K3), mecoprop (O) and clopyralid (O). Flufenacet and terbuthylazine were the only high-use pre-emergent active-ingredient used in wheat (the latter can also be used post-emergent), both were used less in barley crops.

## Cases of resistance by species and risk scoring

The documented cases of herbicide-resistance for medium and high-risk species and all herbicide mode-of-action combinations are shown in Table 3 (a total of 46 medium and high-risk species). High-risk species were the eight grasses *Avena fatua*, *Avena sterilis*, *Digitaria sanguinalis*, *Echinochloa crus-galli*, *Lolium multiflorum*, *Lolium perenne*, *Phalaris minor*, *Poa annua* and eight broadleaf weeds *Amaranthus powellii*, *Chenopodium album*, *Erigeron bonariensis*, *Erigeron sumatrensis*, *Raphanus raphanistrum*, *Senecio vulgaris*, *Solanum nigrum*, *Stellaria media*. Considering our list of weed species known in wheat and barley in New Zealand, we can see that the number of weed species with herbicide resistance documented worldwide varies across HRAC herbicide groups B (34 species), C1 (21), G (16), O (13) and A (12) (Table 3). *Poa annua*, *Echinochloa crus-galli*, *Lolium* spp., *Erigeron sumatrensis*, *Raphanus raphanistrum* and *Avena fatua* have twenty or more recorded cases that occur in five or more unique mode-of-action groups each. We report the low risk weeds, not included in Table 3: *Achillea millefolium*, *Agrostis capillaris*, *Amaranthus deflexus*, *Amsinckia calycina*, *Aphanes arvensis*, *Arrhenatherum elatius*, *Avena barbata*, *Barbarea intermedia*, *Brassica napus*, *Bromus hordeaceus*, *Calandrinia compressa*, *Calandrinia menziesii*, *Cardamine flexuosa*, *Cerastium glomeratum*, *Chenopodiastrum murale*, *Cirsium vulgare*, *Crepis capillaris*, *Crepis setosa*, *Dactylis glomerata*, *Elytrigia repens*, *Erodium cicutarium*, *Erodium moschatum*, *Festuca rubra*, *Fumaria bastardii*, *Fumaria muralis*, *Fumaria officinalis*, *Gamochaeta coarctata*, *Gamochaeta purpurea*, *Geranium molle*, *Leontodon saxatilis*, *Lepidium didymum*, *Lotus pedunculatus*, *Lysimachia arvensis*, *Malva neglecta*, *Malva parviflora*, *Matricaria discoidea*, *Oxalis debilis*, *Oxalis latifolia*, *Phalaris aquatica*, *Phalaris canariensis*, *Ranunculus repens*, *Sherardia arvensis*, *Silene vulgaris*, *Sisymbrium officinale*, *Solanum sarrachoides*, *Stachys arvensis*, *Taraxacum officinale*, *Trifolium pratense*, *Trifolium repens*, *Veronica arvensis*, *Veronica persica*, *Vicia hirsuta*, *Vicia lathyroides*, *Viola arvensis* and *Vulpia myuros*. These weed species are low-risk because they had no reported instances of herbicide-resistance anywhere in the world [4].

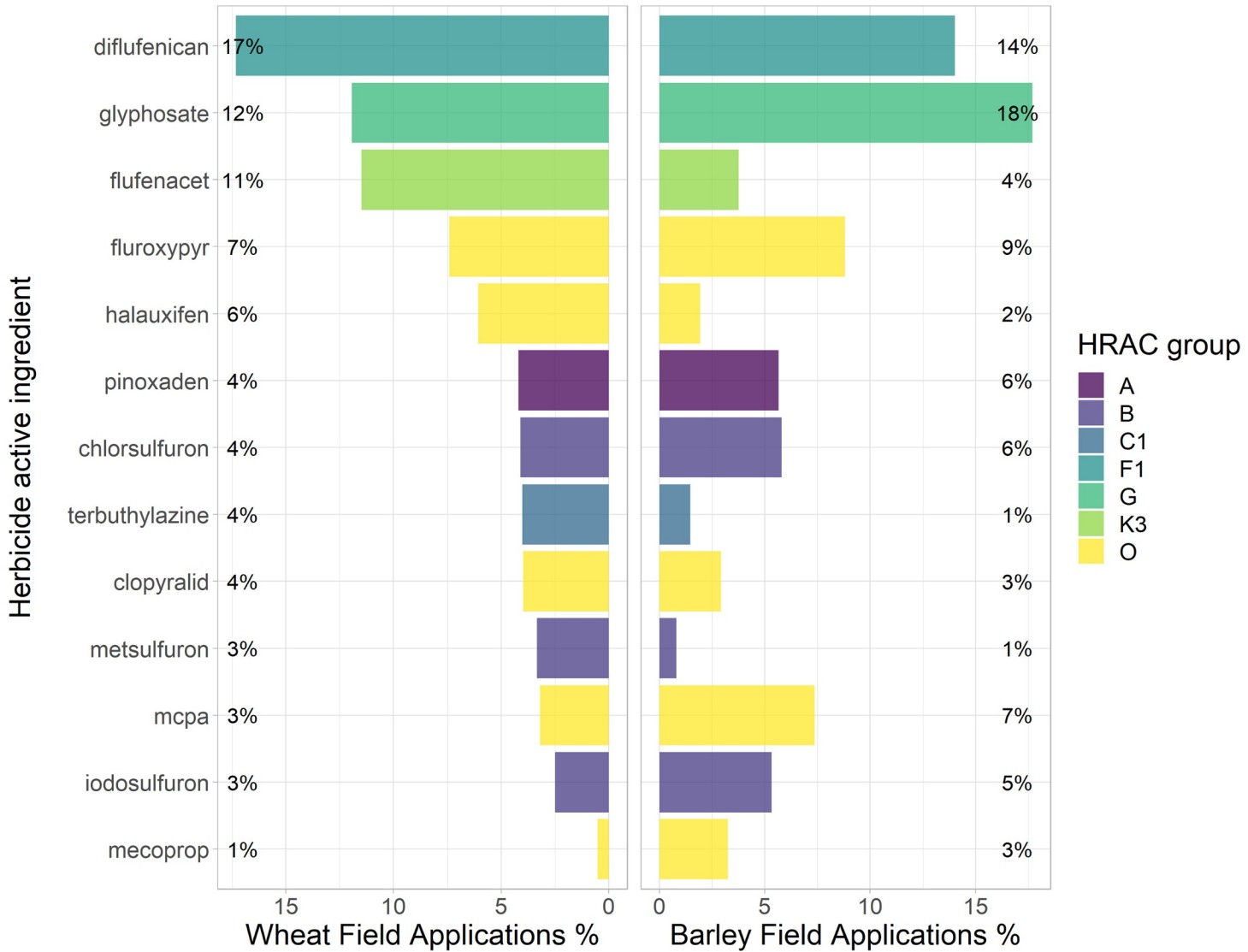

**Fig 1. The ten most common herbicides applied in New Zealand wheat and barley fields.** The ten most used herbicides as a percentage of total application instances documented by farmers in New Zealand wheat and barley fields, here ordered by observations in wheat crops. The herbicides that were not ranked in the top ten for each crop respectively were included here for both of the crops for completeness.

The cumulative risk scores shown in Fig 2 gives a higher score to weeds with resistance to multiple modes-of-action. Because of this, species with high overall risk scores may have relatively few cases of resistance detected overall but their score is inflated by cases of resistance to multiple HRAC modes-of-action. For example *Lolium perenne* (15 cases in 5 modes-of-action) has a slightly lower risk score overall compared to its congener *Lolium multiflorum* (130 cases in 7 modes-of-action). *Chenopodium album* is an example of a relatively low scoring weed that has had 49 cases of resistance documented but within only three HRAC modes-of-action (Fig 2 and Table 3). Risks can be skewed toward certain herbicide modes-of-action, or species. Weeds with the ten highest cumulative scores are *Echinochloa crus-galli*, *Poa annua*, *Lolium multiflorum*, *Erigeron sumatrensis*, *Raphanus raphanistrum*, *Lolium perenne*, *Erigeron bonariensis*, *Avena fatua*, *Avena sterilis*, *Digitaria sanguinalis*. This shows that 7 out of 10 of the high risk species are grass-weeds. Grass-weeds are over-represented in global cases of resistance.

**Table 3. Herbicide resistance risk scores and cases for herbicide mode-of-action groups and species.** For weeds of wheat and barley in New Zealand we document the number of worldwide cases of herbicide resistance within different HRAC [26] groups. Data are sourced from the International Survey of Herbicide-Resistant Weeds [4] and concerns unique modes-of-action within reported cases of herbicide resistance. Species-risk and herbicide-risk are either low (1), medium (2), or high (3). Total cases and species per herbicide mode-of-action across worldwide cases [4] are presented at the bottom of the table.

| Species | Species Risk | A | B | C1 | C2 | C3 | D | E | F1 | F2 | F3 | F4 | G | H | L | K3 | K1 | N | O | Z | Groups | Total Unique Cases |
|---|---|---|---|---|---|---|---|---|---|---|---|---|---|---|---|---|---|---|---|---|---|---|
| **Herbicide Risk** | | 3 | 3 | 3 | 2 | 1 | 2 | 1 | 1 | 1 | 1 | 1 | 3 | 1 | 1 | 1 | 1 | 1 | 2 | 1 | | |
| *Amaranthus powelii* | 3 | 0 | 8 | 8 | 0 | 0 | 0 | 0 | 0 | 0 | 0 | 0 | 0 | 0 | 0 | 0 | 0 | 0 | 0 | 0 | 2 | 16 |
| *Avena fatua** | 3 | 38 | 19 | 0 | 0 | 0 | 0 | 1 | 0 | 0 | 0 | 0 | 0 | 0 | 0 | 1 | 1 | 9 | 0 | 8 | 7 | 77 |
| *Avena sterilis* | 3 | 15 | 9 | 0 | 0 | 0 | 0 | 0 | 0 | 0 | 0 | 0 | 2 | 0 | 0 | 0 | 0 | 0 | 0 | 2 | 4 | 28 |
| *Brassica rapa* | 2 | 0 | 1 | 1 | 0 | 0 | 0 | 0 | 0 | 0 | 0 | 0 | 2 | 0 | 0 | 0 | 0 | 0 | 1 | 0 | 4 | 5 |
| *Bromus catharticus* | 2 | 0 | 0 | 0 | 0 | 0 | 0 | 0 | 0 | 0 | 0 | 0 | 1 | 0 | 0 | 0 | 0 | 0 | 0 | 0 | 1 | 1 |
| *Bromus diandrus* | 2 | 1 | 1 | 0 | 0 | 0 | 0 | 0 | 0 | 0 | 0 | 0 | 1 | 0 | 0 | 0 | 0 | 0 | 0 | 0 | 3 | 3 |
| *Bromus secalinus* | 2 | 0 | 2 | 0 | 0 | 0 | 0 | 0 | 0 | 0 | 0 | 0 | 0 | 0 | 0 | 0 | 0 | 0 | 0 | 0 | 1 | 2 |
| *Bromus sterilis* | 2 | 1 | 2 | 0 | 0 | 0 | 0 | 0 | 0 | 0 | 0 | 0 | 0 | 0 | 0 | 0 | 0 | 0 | 0 | 0 | 2 | 3 |
| *Capsella bursa-pastoris* | 2 | 0 | 6 | 2 | 0 | 0 | 0 | 0 | 0 | 0 | 0 | 0 | 0 | 0 | 0 | 0 | 0 | 0 | 0 | 0 | 2 | 8 |
| *Carduus nutans†* | 2 | 0 | 0 | 0 | 0 | 0 | 0 | 0 | 0 | 0 | 0 | 0 | 0 | 0 | 0 | 0 | 0 | 0 | 1 | 0 | 1 | 1 |
| *Chenopodium album†* | 3 | 0 | 7 | 41 | 0 | 0 | 0 | 0 | 0 | 0 | 0 | 0 | 0 | 0 | 0 | 0 | 0 | 0 | 1 | 0 | 3 | 49 |
| *Cirsium arvense* | 2 | 0 | 0 | 0 | 0 | 0 | 0 | 0 | 0 | 0 | 0 | 0 | 0 | 0 | 0 | 0 | 0 | 2 | 0 | 0 | 1 | 2 |
| *Critesion murinum* | 2 | 3 | 1 | 0 | 0 | 0 | 4 | 0 | 0 | 0 | 0 | 0 | 1 | 0 | 0 | 0 | 0 | 0 | 0 | 0 | 4 | 9 |
| *Convolvulus arvensis* | 2 | 0 | 0 | 0 | 0 | 0 | 1 | 0 | 0 | 0 | 0 | 0 | 0 | 0 | 0 | 0 | 0 | 0 | 0 | 0 | 1 | 1 |
| *Digitaria sanguinalis* | 3 | 7 | 3 | 3 | 0 | 0 | 0 | 0 | 0 | 0 | 0 | 0 | 0 | 0 | 0 | 0 | 0 | 0 | 0 | 0 | 3 | 13 |
| *Echinochloa crus-galli* | 3 | 14 | 16 | 6 | 11 | 0 | 0 | 0 | 0 | 0 | 0 | 1 | 1 | 0 | 2 | 3 | 0 | 2 | 7 | 0 | 10 | 63 |
| *Erigeron bonariensis* | 3 | 0 | 1 | 2 | 0 | 0 | 6 | 0 | 0 | 0 | 0 | 0 | 13 | 0 | 0 | 0 | 0 | 0 | 0 | 0 | 4 | 22 |
| *Erigeron sumatrensis* | 3 | 0 | 7 | 1 | 0 | 0 | 9 | 0 | 0 | 0 | 0 | 0 | 10 | 0 | 0 | 0 | 0 | 1 | 0 | 0 | 5 | 28 |
| *Fallopia convolvulus* | 2 | 0 | 2 | 2 | 0 | 0 | 0 | 0 | 0 | 0 | 0 | 0 | 0 | 0 | 0 | 0 | 0 | 0 | 0 | 0 | 2 | 4 |
| *Fumaria densiflora* | 2 | 0 | 0 | 0 | 0 | 0 | 0 | 0 | 0 | 0 | 0 | 0 | 1 | 0 | 0 | 0 | 1 | 0 | 0 | 0 | 2 | 2 |
| *Galium aparine* | 2 | 0 | 4 | 0 | 0 | 0 | 0 | 0 | 0 | 0 | 0 | 0 | 0 | 0 | 0 | 0 | 0 | 3 | 0 | 0 | 2 | 7 |
| *Lactuca serriola* | 2 | 0 | 5 | 0 | 0 | 0 | 0 | 0 | 0 | 0 | 0 | 0 | 1 | 0 | 0 | 0 | 0 | 1 | 0 | 0 | 3 | 7 |
| *Lamium amplexicaule* | 2 | 0 | 1 | 0 | 0 | 0 | 0 | 0 | 0 | 0 | 0 | 0 | 0 | 0 | 0 | 0 | 0 | 0 | 0 | 0 | 1 | 1 |
| *Lolium multiflorum** | 3 | 55 | 37 | 0 | 0 | 0 | 2 | 0 | 0 | 0 | 1 | 0 | 27 | 3 | 0 | 5 | 0 | 0 | 0 | 0 | 7 | 130 |
| *Lolium perenne** | 3 | 4 | 5 | 0 | 0 | 0 | 0 | 0 | 0 | 0 | 1 | 0 | 4 | 1 | 0 | 0 | 0 | 0 | 0 | 0 | 5 | 15 |
| *Persicaria maculosa†* | 2 | 0 | 1 | 4 | 0 | 0 | 0 | 0 | 0 | 0 | 0 | 0 | 0 | 0 | 0 | 0 | 0 | 0 | 0 | 0 | 2 | 5 |
| *Phalaris minor* | 3 | 9 | 4 | 0 | 0 | 0 | 0 | 0 | 0 | 0 | 0 | 0 | 0 | 0 | 0 | 0 | 0 | 0 | 0 | 0 | 2 | 13 |
| *Phalaris paradoxa* | 2 | 7 | 1 | 1 | 0 | 0 | 0 | 0 | 0 | 0 | 0 | 0 | 0 | 0 | 0 | 0 | 0 | 0 | 0 | 0 | 3 | 9 |
| *Plantago lanceolata* | 2 | 0 | 0 | 0 | 0 | 0 | 0 | 0 | 0 | 0 | 0 | 0 | 1 | 0 | 0 | 0 | 0 | 1 | 0 | 0 | 2 | 2 |
| *Poa annua†* | 3 | 3 | 10 | 19 | 0 | 0 | 2 | 0 | 0 | 0 | 1 | 0 | 7 | 0 | 0 | 0 | 7 | 1 | 4 | 0 | 9 | 54 |
| *Polygonum aviculare* | 2 | 0 | 0 | 1 | 0 | 0 | 0 | 0 | 0 | 0 | 0 | 1 | 0 | 0 | 0 | 0 | 0 | 0 | 0 | 0 | 2 | 2 |
| *Raphanus raphanistrum* | 3 | 0 | 9 | 1 | 0 | 0 | 0 | 0 | 3 | 0 | 0 | 0 | 1 | 0 | 0 | 0 | 0 | 0 | 6 | 0 | 5 | 20 |
| *Rumex acetosella* | 2 | 0 | 0 | 1 | 0 | 0 | 0 | 0 | 0 | 0 | 0 | 0 | 0 | 0 | 0 | 0 | 0 | 0 | 0 | 0 | 1 | 1 |
| *Rumex obtusifolius* | 2 | 0 | 1 | 0 | 0 | 0 | 0 | 0 | 0 | 0 | 0 | 0 | 0 | 0 | 0 | 0 | 0 | 0 | 0 | 0 | 1 | 1 |
| *Senecio vulgaris* | 3 | 0 | 2 | 13 | 0 | 1 | 0 | 0 | 0 | 0 | 0 | 0 | 0 | 0 | 0 | 0 | 0 | 0 | 0 | 0 | 3 | 16 |
| *Silene gallica* | 2 | 0 | 1 | 0 | 0 | 0 | 0 | 0 | 0 | 0 | 0 | 0 | 0 | 0 | 0 | 0 | 0 | 0 | 0 | 0 | 1 | 1 |
| *Solanum americanum†* | 2 | 0 | 0 | 0 | 0 | 0 | 2 | 0 | 0 | 0 | 0 | 0 | 0 | 0 | 0 | 0 | 0 | 0 | 0 | 0 | 1 | 2 |
| *Solanum nigrum†* | 3 | 0 | 0 | 11 | 0 | 0 | 3 | 0 | 0 | 0 | 0 | 0 | 0 | 0 | 0 | 0 | 0 | 0 | 0 | 0 | 2 | 14 |
| *Sonchus asper* | 2 | 0 | 5 | 1 | 0 | 0 | 0 | 0 | 0 | 0 | 0 | 0 | 0 | 0 | 0 | 0 | 0 | 0 | 0 | 0 | 2 | 6 |
| *Sonchus oleraceus* | 2 | 0 | 3 | 0 | 0 | 0 | 0 | 0 | 0 | 0 | 0 | 0 | 1 | 0 | 0 | 0 | 0 | 0 | 2 | 0 | 3 | 6 |
| *Spergula arvensis* | 2 | 0 | 1 | 0 | 0 | 0 | 0 | 0 | 0 | 0 | 0 | 0 | 0 | 0 | 0 | 0 | 0 | 0 | 0 | 0 | 1 | 1 |
| *Stellaria media†* | 3 | 0 | 20 | 1 | 0 | 0 | 0 | 0 | 0 | 0 | 0 | 0 | 0 | 0 | 0 | 0 | 0 | 0 | 2 | 0 | 3 | 23 |
| *Tripleurospermum inodorum* | 2 | 0 | 7 | 0 | 0 | 0 | 0 | 0 | 0 | 0 | 0 | 0 | 0 | 0 | 0 | 0 | 0 | 0 | 0 | 0 | 1 | 7 |
| *Urtica urens* | 2 | 0 | 0 | 1 | 0 | 0 | 0 | 0 | 0 | 0 | 0 | 0 | 0 | 0 | 0 | 0 | 0 | 0 | 0 | 0 | 1 | 1 |

*(Continued)*

**Table 3.** (Continued)

| Species | Species Risk | A | B | C1 | C2 | C3 | D | E | F1 | F2 | F3 | F4 | G | H | L | K3 | K1 | N | O | Z | Groups | Total Unique Cases |
|---|---|---|---|---|---|---|---|---|---|---|---|---|---|---|---|---|---|---|---|---|---|---|
| *Vicia sativa* | 2 | 0 | 1 | 0 | 0 | 0 | 0 | 0 | 0 | 0 | 0 | 0 | 0 | 0 | 0 | 0 | 0 | 0 | 0 | 0 | 1 | 1 |
| *Vulpia bromoides* | 2 | 0 | 0 | 2 | 0 | 0 | 1 | 0 | 0 | 0 | 0 | 0 | 0 | 0 | 0 | 0 | 0 | 0 | 0 | 0 | 2 | 3 |
| **Total cases count** | | 157 | 203 | 122 | 11 | 1 | 30 | 1 | 3 | 0 | 4 | 1 | 74 | 4 | 2 | 9 | 9 | 12 | 32 | 10 | | 683 |
| **Total species count** | | 12 | 34 | 21 | 1 | 1 | 9 | 1 | 1 | 0 | 4 | 1 | 16 | 2 | 1 | 3 | 3 | 3 | 13 | 2 | | 127 |

*Species with herbicide resistance cases detected in New Zealand wheat and barley crops.

†Species with herbicide resistance cases detected in New Zealand, in other crops.

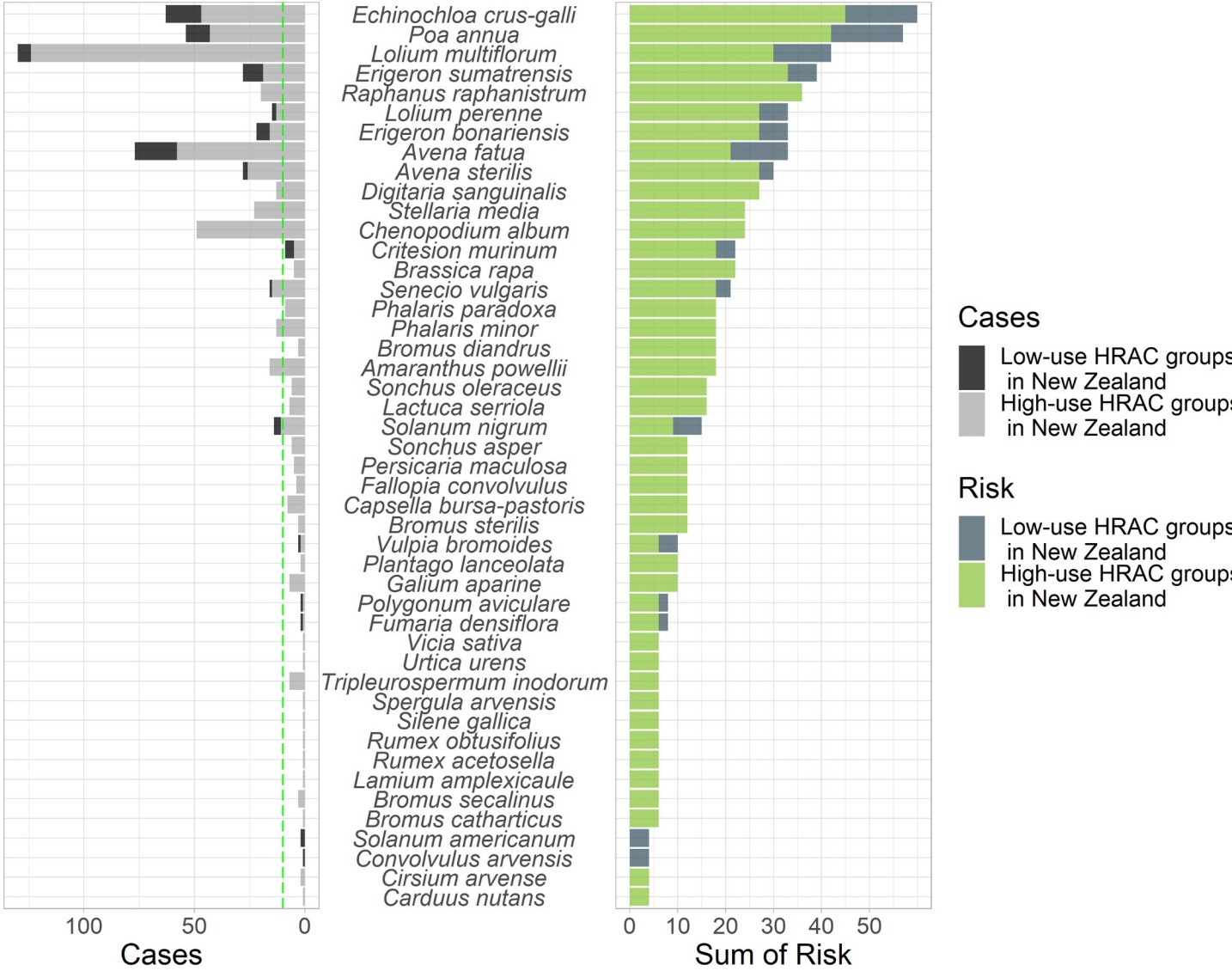

**Fig 2. Number of cases of herbicide resistance (globally) by weed species and cumulative herbicide resistance risk scores.** The Sum of Risk (cumulative risk) scores on the x-axis is the sum of the herbicide-risk × species-risk scores (from Table 3) for each HRAC herbicide mode-of-action group that had documented cases of resistance somewhere in the world. The green dashed line shows the 10 cases needed for a species to be designated high risk (score 3). We distinguished the proportion of the risk and resistance cases that matched with the high-use HRAC mode-of-action groups (A, B, C1, F1, G, K3 & O) used by wheat and barley farmers in New Zealand (light-green for risk, and grey for cases).

## Discussion

We present evidence about the propensity of individual weed species to develop resistance in a curated list (S1 Table) of weed species known to occur in New Zealand wheat and barley fields [22]. The ten highest cumulative risk scores, in order *Echinochloa crus-galli*, *Poa annua*, *Lolium multiflorum*, *Erigeron sumatrensis*, *Raphanus raphanistrum*, *Lolium perenne*, *Erigeron bonariensis*, *Avena fatua*, *Avena sterilis* and *Digitaria sanguinalis*. Because some weeds with moderate-to-high risk scores are widespread, they may be more likely to develop resistance: *Erigeron* spp., *Raphanus raphanistrum*, *Chenopodium album*, *Senecio vulgaris*, *Phalaris* spp., *Bromus diandrus*, *Sonchus oleraceus*, *Solanum nigrum* and *Persicaria maculosa*. Species to species differences in distribution, abundance and phenology (e.g. germination timing) will mean that our risk scores do not capture field level differences in selection pressure from farmer herbicide applications in wheat in barley. As such, emerging grass weeds *Echinochloa crus-galli and Digitaria sanguinalis* are identified as high-risk despite their currently limited distribution. By being aware of all the weed species that are high-risk we should improve detection of resistance cases in future.

A review of the New Zealand literature shows only two reports of resistant species in wheat and barley [7,11]. This contrasts with the high numbers of resistance cases seen worldwide for these crops (77 cases in wheat, and 30 in barley) [4]. One might expect cases to be reported quickly, given that selection for rare mutations that confer resistance is infrequent but instantaneous. But surviving individuals and progeny may take a few years to increase to detectable levels (in a field) under continuous selection pressure [30]. In 2014 *Avena fatua* survivors in wheat and barley fields were shown to be resistant to acetyl coenzyme-A carboxylase (group A, ACCase) [31], and in 2017 ryegrasses (*Lolium perenne* and *L. multiflorum*) in wheat fields were resistant to ACCase and acetolactate synthase targeting herbicides (ALS herbicide, group B) [31,32]. A 1996 report of *Stellaria media* resistance to chlorsulfuron and tribenuron (group B) in an oat crop may indicate elevated risk given the shared agronomic practices between these cereal crops [33]. The resistant weeds previously detected in New Zealand cereals *Lolium* spp., and *Avena* spp. and *Stellaria media*, are therefore likely to continue to be observed. It seems likely that farmers are under-reporting resistance cases perhaps because alternative weed control measures can keep problems manageable. In the absence of a systematic approach, little is known about the spatial and temporal patterns of herbicide resistance development in New Zealand.

Estimating the overall prevalence of herbicide resistance in all the major farming sectors in New Zealand could cost $1–3 million NZD depending on sampling rates [11]. An obvious concern is that detection rates for herbicide-resistant weeds will necessarily underestimate the true rate, given that surveyors may miss individual weed species, resistant plants, or seeds during farm visits [11], also they could miss cases by screening for the wrong herbicides. Surveys have been initiated for the arable sector in New Zealand's Canterbury region where wheat and barley are important crops. They will sample about 20% of ca. 800 arable farms at an estimated cost of ca. $154,000. Given these high survey costs, it is important to take steps to improve detection rates, the high-risk species identified here should be targeted during surveys. Without this list we could be biased toward a smaller number of known problem weeds, such as *Lolium* spp. and *Avena fatua*.

The Moss protocol relies heavily on a herbicide-risk score (rank low, medium and high). We deviated from their approach slightly. This was necessary in part because the number of cases for different herbicide groups has increased since the Moss article was published. A case in point is our decision to include glyphosate as a high-risk herbicide. As recently as 2006 glyphosate was ranked as amongst the least likely to select for resistance [34], and is recognized

as medium-risk by Moss. We ranked glyphosate (group G; Table 1) as high risk because world-wide the number of species showing resistance has increased to 47 cases, similar to group A (48 cases) which is universally regarded as high-risk. Group A herbicides were ranked as high-risk in the Moss protocol. Two cases of glyphosate resistance are known from New Zealand [7,11]. If we had used the 10% threshold groups, A and G would be medium risk and species scores would have changed, but the top five ranked species would have remained the same. We are aware that herbicide-risk is not just a function of the number associated cases of resistance. More complete risk assessments would ideally factor in the herbicide volumes used, years of product use, spatial extent and number of the applications, as well as the abundance of high-risk weeds in the areas treated.

Herbicide usage data in New Zealand have rarely been quantified, and only roughly via indirect sales data numbers and expert elicitation [35,36]. The ProductionWise® system is used to record the on-farm use of chemical inputs and other information. This system (and similar tools) are valuable for ensuring farmer compliance with record-keeping regulations, supporting farmer decision-making, and guiding herbicide resistance prevention efforts. It also serves to capture industry-wide behaviour regarding agrichemical use which is how we have used it in this case. Herbicide resistance risk is influenced by the other crops in the rotation, and temporal and spatial differences in weed composition. The relatively low rates of herbicide resistance detected in the New Zealand arable sector may be a consequence of mixed-crop rotation systems. It is not uncommon to include a 1–3 year pasture rotation, and a complex crop sequence, for example, winter wheat, spring-sown peas or linseed, winter wheat or barley, followed by ryegrass, and oilseed rape and back to winter wheat [37]. Cases of herbicide resistance we observe now only partly reflect current herbicide use and may, in fact, implicate historic selection by herbicides that have fallen out of favour. We think advanced record-keeping tools like ProductionWise® and related decision support systems have real potential to improve outcomes for farmers and scientists.

Within New Zealand wheat and barley fields herbicide use and risk of resistance are greatest within HRAC groups (in order) O, B, G, A and C1. Surprisingly, there have been few cases of resistance to ALS-inhibitors (B) and synthetic auxins (O) documented to date given that they have been the most commonly applied herbicides for more than 20 years [38,39]. Field observations show that broadleaf weeds are rarer in the wheat fields prior to harvest compared to grass weeds, but survivors should be tested for herbicide resistance. Group B herbicides have been implicated in a large number of resistance cases in both broadleaf and grass weeds, including 34 species known to occur in wheat and barley in NZ (Table 3). Groups F1 and K3 have a relatively low risk of developing resistance based on historical occurrences even though they are highly used in New Zealand.

Examining our herbicide use information in the context of published herbicide evolution models can provide important insights. Models based on dryland wheat systems and the weed *Lolium rigidum* [40] showed that resistance rate evolution was not slowed by simple herbicide rotations (i.e. annual with few herbicides). Importantly the use of soil-applied herbicides, particularly trifluralin (group K1), full-rate mixes of herbicides, and complex 8-year long rotations were shown to delay resistance evolution by years, and in some scenarios by decades [40]. There is a chance that resistance cases in soil-applied pre-emergent herbicides are under-reported compared to post-emergent ones (they are harder to test). For now, resistance evolution in those herbicides appears to be slower. Trifluralin did not feature amongst the most common herbicides in the farmer herbicide use data we obtained for 2017 and 2018 (<0.5% of field applications); the only frequently used pre-emergent herbicides were flufenacet and ter-buthylazine. Farmers should be informed about the high-risk species and herbicide combinations, so as to avoid high-risk behaviors, or at least keep an eye out for problems they will

select for. Farmer decision support platforms and research and extension efforts in New Zealand should emphasize mechanical and cultural control measures, the use of soil-applied pre-emergent herbicides, full-strength label-rate herbicide mixtures, crop rotation to utilize herbicides otherwise unavailable and herbicide rotations of key active ingredients to achieve maximum control and to reduce the rate at which herbicide resistance evolves in weed populations.

## Conclusions

A European protocol [20], designed primarily to assist in herbicide authorization procedures, was adapted to assess the risk of herbicide resistance evolving in 101 different weed species known to occur in New Zealand wheat and barley crops. More than half the weeds weeds (55) we assessed were low-risk, 30 were medium-risk and 16 high-risk. The 10 species posing the highest resistance risk were: *Echinochloa crus-galli*, *Poa annua*, *Lolium multiflorum*, *Erigeron sumatrensis*, *Raphanus raphanistrum*, *Lolium perenne*, *Erigeron bonariensis*, *Avena fatua*, *Avena sterilis* and *Digitaria sanguinalis*. To provide important context we also report on herbicide use patterns in New Zealand wheat and barley fields. We are planning extensive surveys in New Zealand to detect new cases of herbicide resistance. The risk assessment outlined in this paper will enable us to prioritise those weeds identified as posing a high resistance risk and, consequently, make better use of available resources. The risk assessment procedure as described in this paper has the potential to be a useful tool for evaluating the risk of herbicide resistance in a wide range of different weed species in other countries too.

## Supporting information

**S1 Table. The full list of weed species considered in our risk assessment for herbicide resistance in New Zealand Wheat and Barley crops.** The list is derived from Bourdôt et al. [22], with additions from literature [23,24], expert knowledge and field observations made in January (late summer) of 2019 and 2020. Common name (in New Zealand) and family name are indicated. Nomenclature follows the New Zealand Flora [25]. </SI_Caption>
(DOCX)

## Acknowledgments

Ian Heap's International survey of herbicide resistant weeds was an important resource for this article. Shona Lamoureaux, Phil Hulme, Trevor James, Hossein Ghanizadeh, Ian Heap and Graeme Bourdôt provided helpful comments on a draft. Implementing the peer reviewer's suggestions improved the article.

## Author Contributions

**Conceptualization:** Zachary Ngow, Christopher E. Buddenhagen.

**Data curation:** Zachary Ngow, Richard J. Chynoweth, Matilda Gunnarsson, Phil Rolston, Christopher E. Buddenhagen.

**Formal analysis:** Zachary Ngow, Christopher E. Buddenhagen.

**Investigation:** Zachary Ngow, Phil Rolston, Christopher E. Buddenhagen.

**Methodology:** Zachary Ngow, Phil Rolston, Christopher E. Buddenhagen.

**Project administration:** Christopher E. Buddenhagen.

**Supervision:** Christopher E. Buddenhagen.

**Validation:** Matilda Gunnarsson, Phil Rolston.

**Visualization:** Christopher E. Buddenhagen.

**Writing – original draft:** Zachary Ngow, Christopher E. Buddenhagen.

**Writing – review & editing:** Zachary Ngow, Richard J. Chynoweth, Matilda Gunnarsson, Phil Rolston, Christopher E. Buddenhagen.

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
