## [Decision Letter · Decision Letter 0]

5 May 2020

PONE-D-20-09801

A herbicide resistance risk assessment for weeds in wheat and barley crops in New Zealand

PLOS ONE

Dear Dr Buddenhagen ,

Thank you for submitting your manuscript to PLOS ONE. After careful consideration, we feel that it has merit but does not fully meet PLOS ONE’s publication criteria as it currently stands. Therefore, we invite you to submit a revised version of the manuscript that addresses the points raised during the review process.

We would appreciate receiving your revised manuscript by Jun 19 2020 11:59PM. To enhance the reproducibility of your results, we recommend that if applicable you deposit your laboratory protocols in protocols.io, where a protocol can be assigned its own identifier (DOI) such that it can be cited independently in the future. For instructions see: http://journals.plos.org/plosone/s/submission-guidelines#loc-laboratory-protocols

We look forward to receiving your revised manuscript.

Kind regards,

Ahmet Uludag, Ph.D.

Academic Editor

PLOS ONE

Additional Editor Comments:

This is a good paper. You will see three reviewer's suggestions. Please prepare a document for me how did you proceed for every suggestion. It can help publishing faster. Good luck.

Journal Requirements:

2. Thank you for stating the following in the Financial Disclosure section:"All the authors worked under the Ministry of Business, Innovation and Employment [grant number C10X1806] to AgResearch Ltd.  https://www.mbie.govt.nz/science-and-technology/science-and-innovation/funding-information-and-opportunities/investment-funds/endeavour-fund/ The funders had no role in study design, data collection and analysis, decision to publish, or preparation of the manuscript."

Thank you for stating the following in the Competing Interests section:"The authors have declared that no competing interests exist."

We note that one or more of the authors are employed by a commercial company:"AgResearch Ltd"

1.Please provide an amended Funding Statement declaring this commercial affiliation, as well as a statement regarding the Role of Funders in your study. If the funding organization did not play a role in the study design, data collection and analysis, decision to publish, or preparation of the manuscript and only provided financial support in the form of authors' salaries and/or research materials, please review your statements relating to the author contributions, and ensure you have specifically and accurately indicated the role(s) that these authors had in your study. You can update author roles in the Author Contributions section of the online submission form.

Reviewers' comments:

Reviewer's Responses to Questions

**Comments to the Author**

1. Is the manuscript technically sound, and do the data support the conclusions?

Reviewer #1: Yes

Reviewer #2: Yes

Reviewer #3: Yes

2. Has the statistical analysis been performed appropriately and rigorously? 

Reviewer #1: N/A

Reviewer #2: N/A

Reviewer #3: No

3. Have the authors made all data underlying the findings in their manuscript fully available?

Reviewer #1: Yes

Reviewer #2: Yes

Reviewer #3: Yes

4. Is the manuscript presented in an intelligible fashion and written in standard English?

Reviewer #1: Yes

Reviewer #2: Yes

Reviewer #3: Yes

5. Review Comments to the Author

Reviewer #1: Because frequency and abundance of weed species in the region have not been used in the scoring, the paper provides a picture of risks due to the current use of herbicides in the region, which is informative for the farmer about the intrinsic risk of an abundant species in its own farm. However, although I agree with intrinsic high risk species as stated on line 280-285, it is pity that the overall risk for the region is not presented.

50-54: There is also an important survey on blackgrass in UK (see Hicks et al Nature Ecology & Evolution 2(3) 2018, DOI: 10.1038/s41559-018-0470-1). Probably other reports are available on surveys developed by national weed research associations in Spain, France (Columa), Italy, etc., including companies, extension service and farmers, and published in local conferences.

142: It can be assumed that weed biology is already included in the inherent species score, but abundance is a key parameter. Abundance includes effects of farming systems: if many individuals still remain in the field, then the risk of developing resistance is higher than if the population is small. A score modifier to account for variation of abundance (e.g. 1 for low density and 3 for more than a million plant/ha, or another scale) could be used. Another factor is the species frequency in the studied region, which doesn’t seem to be taken into account (is the Production Wise platform collecting main troublesome weed species?).

Fig 2: Rather the low and high use HRAC groups I expected to see also the number of cases in New Zealand. I’m not clear with the low versus high use HRAC groups.

Reviewer #2: These comments are exactly the same as in attached file which is in a more convenient format to read and possibly to respond to.

PONE-D-20-09801

A herbicide resistance risk assessment for weeds in

wheat and barley crops in New Zealand

Ngow et al

Line No (as in downloaded pdf).

Abstract: Generally fine but as likely to be most-read part I suggest a few minor changes. Do you need to say ‘wheat and barley’ or would ‘cereal’ suffice? Ditto in title.

I suggest:

15-18 ‘We estimated the risk of selecting for herbicide resistance in 100 weed species known to occur in wheat and barley crops on farms in New Zealand. A protocol was used that accounts for both the risk that different herbicides will select for resistance and each weed’s propensity to develop herbicide resistance based on the number of cases worldwide.’

24 ‘ALS-inhibitors were assessed as posing the greatest risk for more species than any other mode-of-action.

25 I prefer ‘Pre-emergence’ but that may depend on journal style.

26 ‘……. in this class commonly used by…..’ Is this better than ‘favoured’?

Somewhere I think it would be worth stressing in abstract that 7 out of the high-risk 10 are grass-weeds. If space is limiting, I think the last sentence of abstract could go or ‘farmer extension efforts’ incorporated into previous sentence.

Introduction: Good – acceptably concise and relevant. I suggest including some brief (one or two sentences) information on arable cropping in NZ and what proportion of that is accounted for by wheat and barley. Or at least to show that wheat and barley comprise a significant proportion of arable crops – you might even wish to mention that NZ currently holds world record for both barley and wheat yield/ha (I think that is correct). I would!

32 Suggest: ‘……potential losses….’ Losses of 23% don’t actually commonly occur.

41 Clarify ‘cases’. To non-specialist reader it is vague and could mean number of fields or farms – although ‘specialists’ know why that term is appropriate. I would suggest stating number of weed species instead which is 13 according to Heap website country info. That lists 19 ‘cases’ (29 April) although that duplicates some species. If it is 25 cases now (according to your reference), perhaps someone in NZ should provide Ian Heap with updated info. This is important if this paper is published. Likewise amend ‘12’ in line 42 if appropriate.

45 ‘haphazard’ is a bit unfair, although this is a valid point. I would dispute that my sampling and testing over 30 years has been ‘haphazard’! Suggest ‘unsystematic’ as a better word – I would agree with that. End sentence after ‘globally’. Then ‘It reflects…..’

50 comma needed after [9-14]. These 6 refs are all valid, but do you need them all?

53-54 Good points – one issue is that if resistance is perceived to be rare then hard to justify cost. If very widespread then why bother to do a survey? Money could be better spent. Also lack of infrastructure and personnel to conduct surveys may be as important as cost. Also, do surveys have much long-term value? Is a 20-year old survey of much value now – perhaps only as a benchmark? Perhaps money better spent on more ‘durable’ studies? Not saying that you need to consider these aspects here – perhaps in discussion as risk assessment relevant to survey priorities.

59 Suggest: ‘…….and their prior record of resistance in cereal fields elsewhere in the world.’

62-64 This is fine and a good succinct sentence, but I wonder if it is worth emphasising more that risk assessment includes not only herbicide risk but also weed species risk for non-specialist readers, as this is not particularly intuitive? So, could read: ‘They present a quantitative risk matrix using both herbicide-risk (some herbicides pose a higher risk than others) and species-risk (some weed species are more resistance-prone than others), with an optional score modifier designed to account for agronomic management practices that may reduce the risk.’

67 Change: ‘Individual field and farm scale risk is not assessed as this requires detailed information on past herbicide use, including timing etc…..’ (This covers what has been applied - surely most important factor?)

Materials and methods: Good – makes it clear what was done and why slightly different approaches to published protocol were adopted

76 Suggest: ‘Most grasses and some …..’ The wonders of Google mean that I can see, within 60 seconds, that two grass weed species were identified at species level in the paper cited…..

83 Suggest: ‘….legacy herbicide mode of action (MoA) groups……’

88-89 Suggest: ‘….legacy HRAC MoA group [24], with risk scores of 1, 2 or 3 given for low, medium or high risk respectively.’

96 Suggest: ‘…..scored as ‘1’.’ (‘One’ is a bit ambiguous).

98 Suggest: ‘Most recent….’

108-109 This is slightly confusing: ‘The most used herbicides for each crop were characterized by weighting active ingredient amounts and the number of fields they were applied to.’ It almost implies total a.i. weight was used. Not sure this sentence is needed.

110 Why taxon? Species better?

112 Suggest: ‘…..HRAC MoA group…...’ And I suggest elsewhere throughout paper.

113 Suggest: ‘….herbicide type….

114 Suggest: ‘……the global number of resistance cases….’

115 Suggest: ‘To obtain the ‘high’, ‘medium’ and ‘low’ risk scores as used in the Moss protocol [17],…..’

117-118 This sentence is slightly confusing and maybe could be improved. Is this correct? ‘We assessed overall species-risk as the sum of the herbicide-risk multiplied by the “inherent” species-risk [17] combined for all relevant HRAC MoA herbicide groups, but only…….’ (I think ‘once’ is confusing). The example you give below is useful.

120 Suggest: ’……weed species….’

135 Suggest: ‘……to determine species-risk scores based on the number of cases of resistance worldwide, but we think this……’

137 Suggest: ‘ ….45 ‘high’ and ‘medium’ resistance risk species, many more than Moss et al……….’

139 Suggest: ‘The Moss protocol also used score modifiers that take into account resistance management practices including the use of non-chemical control measures’ .

139-142 Suggest: ‘We did not use the score modifiers since these vary from field to field and our objective was slightly different. We acknowledge that actual risk of resistance development is determined mostly by the frequency and type of herbicide applied (selection pressure) interacting with characteristics of weed biology, distribution and abundance [5].’

Results: Generally good and concise. One key paragraph needs improvement to improve clarity

145 Suggest: ‘An additional 31 species were added…..’

147 Suggest: ‘ ….resulting in a total of 100 weed species for consideration.’

155 Suggest: Suggest: ‘This resembles the table of Moss et al., [17] with…’

Table 1 Use of ‘taxa’ is odd. Why not ‘species’ which is what is used on Heap’s website? Taxa is a rather more general term and I cannot see any justification for use here. Would it also be useful to make the herbicide example the most commonly used ai for the group in NZ? Or if this has been done, state in title.

171-172 Clarify if glyphosate used in-crop (for desiccation) or pre-sowing or both. Presumably both, but some comment about balance of use would be useful if only to stress that glyphosate use is very different to selective herbicides – non-specialists might assume use is primarily in GM crops, as in some other countries. Either here or earlier in M&M.

Suggest: ‘ …….in barley (18%) compared with wheat (12%); conversely, farmers………’

174 Again, ‘HRAC MoA categories’

Table 2 Suggest last 10 categories are simply summarised to save space and some statement added within table. e.g. ‘The following 10 HRAC MoA groups each accounted for less than 1% of herbicide applications, N, F4, K1, I, H, F3, Z, L, F2 & K2.’

181 I don’t really see the point of stating actual % for barley when you haven’t for wheat. Why not simply add % values to each of the bars in Fig 1. You could then say a bit about relative use of some individual herbicides in wheat and barley. Somewhere you should give some indication of relative area sown with wheat and barley – in introduction or M&M.

183 Suggest: ‘Flufenacet and terbuthylazine were the only herbicides used on a significant area pre-emergence (the latter can also be used post-emergence), both used less in barley crops.’

185 Suggest: ‘Fig 1. The ten most commonly applied herbicides in New Zealand wheat and barley fields.’

190 Why taxon? Surely ‘species’ is better?

193-194 Say eight grasses and eight broad-leafed weeds.

201 Suggest: ‘….five or more unique mode-of-action groups each’ What does>20 mean? Clarify.

202-215 This paragraph covers the key output of this paper but is confusing and really needs a thorough re-write. Expand if necessary, to clarify results. Fig 2 contains a lot of information. I did wonder how much the ‘cases’ adds to this but I can see that it is relevant. Some definition of ‘cases’ is needed, as mentioned earlier. This is tricky as not easy to explain in a few words, but if this is explained in the M&M then no need to explain again, except perhaps to say refer to M&M. Should be made clear in Figure that cases is worldwide, as might be interpreted by casual reader as in NZ.

212-215 I would make the point that 7 out of 10 are grass-weeds. Grass-weeds are over-represented in global cases of resistance. Some comment on this in discussion maybe.

Discussion: The paper would benefit from a more concise discussion with improved, more logical structure. Content is OK but lacking in focus and a bit rambling. I suggest reducing it to about 60% max of current length by heavily reducing some paragraphs or omitting altogether. The focus should be on the lessons and implications from what you have presented in the paper. Some of Discussion, while valid, reduces the impact of the paper rather than enhances it.

226-235 Better if this is later in Discussion.

236-261 Reduce this paragraph very substantially. Some of this could be in introduction.

There appear to be some contradictions quoted: ‘77 cases in wheat and 30 cases in barley [4].’ ‘ A review of the New Zealand literature shows only two reports of resistant species in wheat and barley [7,8].’ ‘Cases’ is confusing as different, I think, to previous use of term.

262-287 Suggest this starts the Discussion as seems more logical.

288-306 Again lacks focus. Glyphosate comments relevant. But to make some suggestion but then say ‘Estimations of these other factors can be made, but are likely to be inaccurate.’ Is a case of ‘shooting yourself in the foot.’ Ditto ‘Using Beckie’s as our herbicide-risk ranking would produce different risk-scores.’ Omit.

Other paragraphs – consider what content really adds to paper and what can be omitted.

One aspect that seems missing, is some idea of the crop rotations used on NZ farms and the impact this might have. This focus in this paper is on wheat and barley but the frequency of growing these crops and consequent herbicide use will surely impact on the resistance risk? How commonly is arable cropping rotated with grass? Are all-arable farms common compared with mixed farms? This may explain why herbicide resistance is relatively uncommon in NZ. Also relevant to risk at individual field level which lends itself nicely to comments about where you should set priorities for resistance monitoring, management and farmer KT. No need to go into great detail, but is relevant.

Conclusions: Generally fine but suggest the following.

343-346 Suggest: ‘A European protocol [17], designed primarily to assist in herbicide authorization procedures, was adapted to assess the risk of herbicide resistance evolving in 100 different weed species known to occur in New Zealand wheat and barley crops. More than half the weeds (55) we assessed were low-risk, 29 were medium-risk and 16 high-risk. The 10 species posing the highest resistance risk were: etc etc’

349-351 Final two sentences could usefully be a bit ‘punchier’. ’Suggest: ‘We are planning extensive surveys in New Zealand to detect new cases of herbicide resistance. The risk assessment outlined in this paper will enable us to prioritise those weeds identified as posing a high resistance risk and, consequently, make better use of available resources. The risk assessment procedure as described in this paper has the potential to be a very useful tool for evaluating the risk of herbicide resistance in a wide range of different weed species in other countries too’.

Reviewer #3: Reviewer (YASEEN KHALIL):

Thanks for allowing me to review the manuscript entitle “A herbicide resistance risk assessment for weeds in wheat and barley crops in New Zealand”. This is a very well written and informative manuscript. Authors have carried out a systematic study to determine the quantitative risk matrix using ranked herbicide-risk and species-risk with optional score modifier designed to account for agronomic management practices that may reduce the risk. The researchers took advantage of a unique data set about herbicide use in wheat and barley fields in New Zealand to place their risk assessment into context, and construct a framework for herbicide resistance surveys and extension efforts in the New Zealand cropping industry. This study would be provided a very good message for the growers about the herbicide resistance risk assessment on the industry-wide scale, but not on the field and farm scale. I am in favor of publishing this manuscript to the PLOS ONE Journal. I have very few comments.

GENERAL COMMENTS:

The manuscript represents a review article instead of research article, as it is been mentioned in the manuscript draft, in an important area of herbicide research and was conducted with reasonable methods. In addition, the manuscript is written well, with areas for improvement in several instances.

Keywords should be main words that are not included in the title and the abstract.

The abstract is well structured. I suggest mentioning the scientific name of the weeds including the name of classifiers for the first time.

The introduction and methodology of the experiments seem to be satisfactory. However, in the methods section, the authors need to explain how they did the statistical analyses and data presentation.

SPECIFIC COMMENTS:

L20-22 Add the classifiers to the scientific names. Alternatively, you may add the classifiers to the scientific names in the Materials and methods section and after that no need for it to be mentioned.

L20-22 Add hyphen “herbicide-resistant”

L45 Add the “it reflects the varying”

L52 Add the classifier to the scientific name “Alopecurus myosuroides ”.

L20-22 Add hyphen “farm-scale”

L74-76 What about the other 25% of the wheat and barley production regions in New Zealand. I would prefer to stick to the mentioned region and not extrapolate to the whole country of New Zealand.

L78 It will be very useful to mention the regions of the studies conducted by the mentioned researcher’s literature [21, 22].

L79 It is recommended to follow the journal guidelines in this regards instead (Species nomenclature).

L91-92 What is the logic and the rationale behind this decision? Worldwide, Group A herbicides are the most vulnerable group in terms of weed resistance evolving

L122 Add the classifier to the scientific name “Chenopodium album”.

L140 Add the “that the actual”

L142 Add comma “distribution, and abundance”

L147 Add of “total of 100 weeds”

L150 I would recommend adding the S1 Table to the text instead of having it as a supporting file.

L152 I would recommend adding the S2 Table to the text instead of having it as a supporting file.

L179 Add the “ranked by the wheat”

L203 Add comma “modes-of-action, and”

L234 Add comma “Without this list, we”

L234 replace are with a “As a result”

L240 Add comma “cases in wheat, and”

L246 Add s “indicates elevated”

L247 Add comma “surprising when”

L256 delete “Clearly”

L275, 281, 284 Add hyphens to “spring-sown cereals”, “on-field”, “high-risk”

L289“because of the number”

L290 have instead of has “groups have increased”

L296“because of worldwide”

L308 Surprisingly, there have

L324 Add commas “may, in fact, implicate”

6. PLOS authors have the option to publish the peer review history of their article (what does this mean?). If published, this will include your full peer review and any attached files.

Reviewer #1: No

Reviewer #2: Yes: Dr Stephen Moss

Reviewer #3: No

---

## [Author Response · Author response to Decision Letter 0]

18 May 2020

Reviewer #1: 

Because frequency and abundance of weed species in the region have not been used in the scoring, the paper provides a picture of risks due to the current use of herbicides in the re-gion, which is informative for the farmer about the intrinsic risk of an abundant species in its own farm. However, although I agree with intrinsic high risk species as stated on line 280-285, it is pity that the overall risk for the region is not presented.

The regional risk is assessed! We use a published account of weeds, as well as ex-pert knowledge to identify problem weeds in wheat and barley. The risk assess-ment considers the propensity of those weeds to develop resistance. It uses the most important factors, the crop specific region wide weed flora and region wide data about herbicide use. The selection pressure comes from the herbicides used and so captures the most important dimensions of the regional risk.

L50-54 There is also an important survey on blackgrass in UK (see Hicks et al Nature Ecology & Evolution 2(3) 2018, DOI: 10.1038/s41559-018-0470-1). Probably other reports are availa-ble on surveys developed by national weed research associations in Spain, France (Columa), Italy, etc., including companies, extension service and farmers, and published in local con-ferences.

That is another example of the “focus on one or two problematic species in a given crop” issue we highlight, and therefore does not need to be added.

L142 It can be assumed that weed biology is already included in the inherent species score, but abundance is a key parameter. Abundance includes effects of farming systems: if many individuals still remain in the field, then the risk of developing resistance is higher than if the population is small. A score modifier to account for variation of abundance (e.g. 1 for low density and 3 for more than a million plant/ha, or another scale) could be used. Another factor is the species frequency in the studied region, which doesn’t seem to be taken into account (is the Production Wise platform collecting main troublesome weed species?).

While that is true, regional abundance and density data do not exist to be able to implement that. We acknowledge that distribution, abundance and phenology are important #(L272). ProductionWise does not gather any information on weeds. Be-cause we identified high-risk species and herbicides the data are useful as a region-al assessment. Farmers and experts can now interpret the observed weed abun-dance with an eye to the likely risk of resistance.

Fig 2 Rather the low and high use HRAC groups I expected to see also the number of cases in New Zealand. I’m not clear with the low versus high use HRAC groups.

We emphasize high use herbicides in our studied regions, because that selection pressure drives the evolution of resistance. We mention in the introduction that in New Zealand, there are only 12 instances of herbicide resistance in arable crops #(L41-43) and in the discussion delve further into New Zealand HR weeds (L238-244). The high and low usage herbicide groups come from ProductionWise data, which can be found on Table 2. The basis of deciding which groups are ‘high use in New Zealand’ is on L106-108. Cases of resistance have been indicated on Table 3.

Reviewer #2 Dr Stephen Moss: 

Abstract: Generally fine but as likely to be most-read part I suggest a few minor changes. Do you need to say ‘wheat and barley’ or would ‘cereal’ suffice? Ditto in title.

We think wheat and barley is a more accurate statement for our case since the herbicide data focuses on that. Wheat and barley is more widely grown than oats, and depending on the reader, corn would also be consid-ered to be a cereal…

 I suggest:

15-18 ‘We estimated the risk of selecting for herbicide resistance in 100 weed species known to occur in wheat and barley crops on farms in New Zealand. A protocol was used that accounts for both the risk that different herbicides will select for resistance and each weed’s propensity to develop herbicide resistance based on the number of cases worldwide.’ Done

24 ‘ALS-inhibitors were assessed as posing the greatest risk for more species than any other mode-of-action. Done

25 I prefer ‘Pre-emergence’ but that may depend on journal style. Done

26 ‘……. in this class commonly used by…..’ Is this better than ‘favoured’? Done

 Somewhere I think it would be worth stressing in abstract that 7 out of the high-risk 10 are grass-weeds. If space is limiting, I think the last sentence of abstract could go or ‘farmer extension efforts’ incorporated into previous sentence. Done

Introduction: Good – acceptably concise and relevant. I suggest including some brief (one or two sentences) information on arable cropping in NZ and what proportion of that is accounted for by wheat and barley. Or at least to show that wheat and barley comprise a significant proportion of arable crops – you might even wish to mention that NZ currently holds world record for both barley and wheat yield/ha (I think that is correct). I would! Done

32 Suggest: ‘……potential losses….’ Losses of 23% don’t actually commonly occur. Done

41 Clarify ‘cases’. To non-specialist reader it is vague and could mean number of fields or farms – although ‘specialists’ know why that term is appropriate. I would suggest stating number of weed species instead which is 13 according to Heap website country info. That lists 19 ‘cases’ (29 April) although that dupli-cates some species. If it is 25 cases now (according to your reference), perhaps someone in NZ should provide Ian Heap with updated info. This is important if this paper is published. Likewise amend ‘12’ in line 42 if appropriate. Done re-worded to reflect species numbers in New Zealand, and documented instances of resistance, to distinguish from the usage of cases in the Heap database.

45 ‘haphazard’ is a bit unfair, although this is a valid point. I would dispute that my sampling and testing over 30 years has been ‘haphazard’! Suggest ‘unsystemat-ic’ as a better word – I would agree with that. End sentence after ‘globally’. Then ‘It reflects…..’ Done

50 comma needed after [9-14]. These 6 refs are all valid, but do you need them all? They are valid, focus on systematic efforts to detect any and all weed re-sistant cases. Also, they could be hard to find for other researchers working on a similar project. Leaving as is.

53-54 Good points – one issue is that if resistance is perceived to be rare then hard to justify cost. If very widespread then why bother to do a survey? Money could be better spent. Also lack of infrastructure and personnel to conduct surveys may be as important as cost. Also, do surveys have much long-term value? Is a 20-year old survey of much value now – perhaps only as a benchmark? Perhaps money better spent on more ‘durable’ studies? Not saying that you need to con-sider these aspects here – perhaps in discussion as risk assessment relevant to survey priorities. Good points thanks for the insightful commentary.

59 Suggest: ‘…….and their prior record of resistance in cereal fields elsewhere in the world.’ We specifically looked at wheat and barley. It’s an important dis-tinction.

62-64 This is fine and a good succinct sentence, but I wonder if it is worth emphasising more that risk assessment includes not only herbicide risk but also weed species risk for non-specialist readers, as this is not particularly intuitive? So, could read: ‘They present a quantitative risk matrix using both herbicide-risk (some herbicides pose a higher risk than others) and species-risk (some weed species are more resistance-prone than others), with an optional score modifier de-signed to account for agronomic management practices that may reduce the risk.’ Done

67 Change: ‘Individual field and farm scale risk is not assessed as this requires de-tailed information on past herbicide use, including timing etc…..’ (This covers what has been applied - surely most important factor?) We changed the para-graph to emphasize importance of our herbicide application data set: “. This risk assessment is on an industry-wide scale informed by anonymized herbicide application data from wheat and barley fields. Risks were not assessed at the scale of individual farms and fields, this requires detailed information about herbicide timing, mixtures and rotations, and their interactions with weed bi-ology, crop rotations and other cultural practices. All the high-risk weeds iden-tified here should be targeted in surveys designed to detect herbicide-resistant weeds.

Materials and methods: Good – makes it clear what was done and why slightly differ-ent approaches to published protocol were adopted

76 Suggest: ‘Most grasses and some …..’ The wonders of Google mean that I can see, within 60 seconds, that two grass weed species were identified at species level in the paper cited….. Done

83 Suggest: ‘….legacy herbicide mode of action (MoA) groups……’ Done

88-89 Suggest: ‘….legacy HRAC MoA group [24], with risk scores of 1, 2 or 3 given for low, medium or high risk respectively.’ Done

96 Suggest: ‘…..scored as ‘1’.’ (‘One’ is a bit ambiguous). Done

98 Suggest: ‘Most recent….’ Done but, “The most recent…”

108-109 This is slightly confusing: ‘The most used herbicides for each crop were charac-terized by weighting active ingredient amounts and the number of fields they were applied to.’ It almost implies total a.i. weight was used. Not sure this sen-tence is needed. Deleted the sentence.

110 Why taxon? Species better? Changed to species.

 112 Suggest: ‘…..HRAC MoA group…...’ And I suggest elsewhere throughout paper. Done

113 Suggest: ‘….herbicide type…. Done

114 Suggest: ‘……the global number of resistance cases….’ Done

115 Suggest: ‘To obtain the ‘high’, ‘medium’ and ‘low’ risk scores as used in the Moss protocol [17],…..’ Done

117-118 This sentence is slightly confusing and maybe could be improved. Is this correct? ‘We assessed overall species-risk as the sum of the herbicide-risk multiplied by the “inherent” species-risk [17] combined for all relevant HRAC MoA herbicide groups, but only…….’ (I think ‘once’ is confusing). The example you give below is useful. Done

120 Suggest: ’……weed species….’ Done

135 Suggest: ‘……to determine species-risk scores based on the number of cases of resistance worldwide, but we think this……’ Done

137 Suggest: ‘ ….45 ‘high’ and ‘medium’ resistance risk species, many more than Moss et al……….’ Done

139 Suggest: ‘The Moss protocol also used score modifiers that take into account re-sistance management practices including the use of non-chemical control measures’ . Done

139-142 Suggest: ‘We did not use the score modifiers since these vary from field to field and our objective was slightly different. We acknowledge that actual risk of re-sistance development is determined mostly by the frequency and type of herbi-cide applied (selection pressure) interacting with characteristics of weed biolo-gy, distribution and abundance [5].’ Done

Results: Generally good and concise. One key paragraph needs improvement to im-prove clarity

145 Suggest: ‘An additional 31 species were added…..’ Done

147 Suggest: ‘ ….resulting in a total of 100 weed species for consideration.’ Done

155 Suggest: Suggest: ‘This resembles the table of Moss et al., [17] with…’ Done

Table 1 Use of ‘taxa’ is odd. Why not ‘species’ which is what is used on Heap’s website? Taxa is a rather more general term and I cannot see any justification for use here. Would it also be useful to make the herbicide example the most common-ly used ai for the group in NZ? Or if this has been done, state in title. Done

171-172 Clarify if glyphosate used in-crop (for desiccation) or pre-sowing or both. Pre-sumably both, but some comment about balance of use would be useful if only to stress that glyphosate use is very different to selective herbicides – non-specialists might assume use is primarily in GM crops, as in some other coun-tries. Either here or earlier in M&M. Done “Glyphosate (MoA group G) is most-ly used (>95%) used pre-sowing of the cereal crops, for termination of the pre-vious crop or pre-establishment weed control. It is very rarely used as crop pre-harvest desiccant.”

 Suggest: ‘ …….in barley (18%) compared with wheat (12%); conversely, farm-ers………’ Done 

174 Again, ‘HRAC MoA categories’ Done

Table 2 Suggest last 10 categories are simply summarised to save space and some statement added within table. e.g. ‘The following 10 HRAC MoA groups each ac-counted for less than 1% of herbicide applications, N, F4, K1, I, H, F3, Z, L, F2 & K2.’ Done

181 I don’t really see the point of stating actual % for barley when you haven’t for wheat. Why not simply add % values to each of the bars in Fig 1. You could then say a bit about relative use of some individual herbicides in wheat and barley. Somewhere you should give some indication of relative area sown with wheat and barley – in introduction or M&M. We added the percent values to the graph and clarified a comment about relative use under Table 2. The areas sown for wheat and barley are now in the introduction.

183 Suggest: ‘Flufenacet and terbuthylazine were the only herbicides used on a sig-nificant area pre-emergence (the latter can also be used post-emergence), both used less in barley crops.’

185 Suggest: ‘Fig 1. The ten most commonly applied herbicides in New Zealand wheat and barley fields.’ Done

190 Why taxon? Surely ‘species’ is better? Done

193-194 Say eight grasses and eight broad-leafed weeds. Done 

201 Suggest: ‘….five or more unique mode-of-action groups each’ What does>20 mean? Clarify. Done

202-215 This paragraph covers the key output of this paper but is confusing and really needs a thorough re-write. Expand if necessary, to clarify results. Fig 2 contains a lot of information. I did wonder how much the ‘cases’ adds to this but I can see that it is relevant. Some definition of ‘cases’ is needed, as mentioned earlier. This is tricky as not easy to explain in a few words, but if this is explained in the M&M then no need to explain again, except perhaps to say refer to M&M. Should be made clear in Figure that cases is worldwide, as might be interpreted by casual reader as in NZ. Done Figure 2 clearly mentions global cases. Also added the following to the methods: “Cases are defined by the International Sur-vey of Herbicide Resistant Weeds as unique combinations of weed species and HRAC herbicide mode-of-action (species x site of action).” 

212-215 I would make the point that 7 out of 10 are grass-weeds. Grass-weeds are over-represented in global cases of resistance. Some comment on this in discussion maybe. Done

 Discussion: The paper would benefit from a more concise discussion with improved, more logical structure. Content is OK but lacking in focus and a bit ram-bling. I suggest reducing it to about 60% max of current length by heavily reducing some paragraphs or omitting altogether. The focus should be on the lessons and implications from what you have presented in the paper. Some of Discussion, while valid, reduces the impact of the paper rather than enhances it. 

226-235 Better if this is later in Discussion. Done

236-261 Reduce this paragraph very substantially. Some of this could be in introduction.

 There appear to be some contradictions quoted: ‘77 cases in wheat and 30 cases in barley [4].’ ‘ A review of the New Zealand literature shows only two reports of resistant species in wheat and barley [7,8].’ ‘Cases’ is confusing as different, I think, to previous use of term. Done

262-287 Suggest this starts the Discussion as seems more logical. Done 

288-306 Again lacks focus. Glyphosate comments relevant. But to make some sugges-tion but then say ‘Estimations of these other factors can be made, but are likely to be inaccurate.’ Is a case of ‘shooting yourself in the foot.’ Ditto ‘Using Beck-ie’s as our herbicide-risk ranking would produce different risk-scores.’ Omit. Several sentences deleted and reworded to avoid “shooting ourselves in the foot” but keeping the main ideas as a reasonable caveat around interpretation of our results.

 Other paragraphs – consider what content really adds to paper and what can be omitted.

 One aspect that seems missing, is some idea of the crop rotations used on NZ farms and the impact this might have. This focus in this paper is on wheat and barley but the frequency of growing these crops and consequent herbicide use will surely impact on the resistance risk? How commonly is arable cropping ro-tated with grass? Are all-arable farms common compared with mixed farms? This may explain why herbicide resistance is relatively uncommon in NZ. Also relevant to risk at individual field level which lends itself nicely to comments about where you should set priorities for resistance monitoring, management and farmer KT. No need to go into great detail, but is relevant. We have includ-ed some text about rotations.

Conclusions: Generally fine but suggest the following.

343-346 Suggest: ‘A European protocol [17], designed primarily to assist in herbicide authorization procedures, was adapted to assess the risk of herbicide re-sistance evolving in 100 different weed species known to occur in New Zealand wheat and barley crops. More than half the weeds (55) we assessed were low-risk, 29 were medium-risk and 16 high-risk. The 10 species posing the highest resistance risk were: etc etc’ Done

349-351 Final two sentences could usefully be a bit ‘punchier’. ’Suggest: ‘We are plan-ning extensive surveys in New Zealand to detect new cases of herbicide re-sistance. The risk assessment outlined in this paper will enable us to prioritise those weeds identified as posing a high resistance risk and, consequently, make better use of available resources. The risk assessment procedure as described in this paper has the potential to be a very useful tool for evaluating the risk of herbicide resistance in a wide range of different weed species in other coun-tries too’. Done

Reviewer #3 (YASEEN KHALIL):

The introduction and methodology of the experiments seem to be satisfactory. 

However, in the methods section, the authors need to explain how they did the statistical analyses and data presentation.

We do not carry out any statistical tests. All our data is descriptive, and are visual-ized using graphs . We added a sentence to the last line of the methods section about the graphs. 

SPECIFIC COMMENTS:

L20-22 Add the classifiers to the scientific names. Alternatively, you may add the classifiers to the scientific names in the Materials and methods section and after that no need for it to be mentioned.

 We think the reviewer is referring to the specific binomial author authorities. We examined other abstracts in PLOS ONE and do not see the these being used for ab-stracts unless it is specifically about a taxonomic treatment.

L20-22 Add hyphen “herbicide-resistant” Done but line 27

L45 Add the “it reflects the varying” Done

L52 Add the classifier to the scientific name “Alopecurus myosuroides ”. Done and also for Avena fatua

L20-22 Add hyphen “farm-scale” We added this at line 68 though which was the first men-tion

L74-76 What about the other 25% of the wheat and barley production regions in New Zea-land. I would prefer to stick to the mentioned region and not extrapolate to the whole coun-try of New Zealand.

We intend the weed list to capture our best estimate of the weeds known to occur in wheat and barley fields in New Zealand. We added a new sentence to explain: We expanded the weed species list to include species known to occur in wheat and barley fields in the wider New Zealand context.

L78 It will be very useful to mention the regions of the studies conducted by the mentioned researcher’s literature [21, 22].

We disagree. We checked the regions studied in those articles and knowing them adds lit-tle insight. Also, only a few species were added from these.

 L79 It is recommended to follow the journal guidelines in this regards instead (Species no-menclature). Taxonomic authorities added at first mention.

L91-92 What is the logic and the rationale behind this decision? Worldwide, Group A herbi-cides are the most vulnerable group in terms of weed resistance evolving 

We do classify Group A herbicides as high risk. We include a bit more detail here and consider the issues in the discussion. It now reads: With group A having 48 cas-es and group G herbicides having 47 cases we chose to place the two groups in the same risk category, with a difference in the numbers of cases of just worldwide we believe they are indistinguishable from the data. The alternative is to use the same threshold as in the Moss protocol, but this would result in group A and G being me-dium risk, which fails to capture the high-risk status of group A herbicides.

L122 Add the classifier to the scientific name “Chenopodium album”. Done

L140 Add the “that the actual” Done

L142 Add comma “distribution, and abundance” Done

L147 Add of “total of 100 weeds” Done, but it is now 101 weeds.

L150 I would recommend adding the S1 Table to the text instead of having it as a supporting file. Done

L152 I would recommend adding the S2 Table to the text instead of having it as a supporting file. Done We think this one should stay where it is because of its length.

L179 Add the “ranked by the wheat” Done

L203 Add comma “modes-of-action, and” Done

L234 Add comma “Without this list, we” Done

L234 replace are with a “As a result” Done

L240 Add comma “cases in wheat, and” Done

L246 Add s “indicates elevated” Not done. “may indicate” is correct

L247 Add comma “surprising when” Not done. Comma is correct

L256 delete “Clearly” Done

L275, 281, 284 Add hyphens to “spring-sown cereals” Done, “on-field” Not done the acting on field experience” refers to experience gained in the field, “high-risk” Done

L289“because of the number” Not done. This makes sense “This was necessary in part be-cause the number of cases for different herbicide groups has increased since the Moss arti-cle was published.”

L290 have instead of has “groups have increased” Not done. This makes sense “This was necessary in part because the number of cases for different herbicide groups has increased since the Moss article was published.”

L296“because of worldwide” Not done, it makes sense the way it is.

L308 Surprisingly, there have Done

L324 Add commas “may, in fact, implicate” Done

---

## [Editor Report · Decision Letter 1]

3 Jun 2020

A herbicide resistance risk assessment for weeds in wheat and barley crops in New Zealand

PONE-D-20-09801R1

Dear Dr. Chris Evan Buddenhagen,

We’re pleased to inform you that your manuscript has been judged scientifically suitable for publication and will be formally accepted for publication once it meets all outstanding technical requirements and edits suggested by academic editor..

Kind regards,

Ahmet Uludag, Ph.D.

Academic Editor

PLOS ONE

Additional Editor Comments (optional):

I went through all over the document. I have some suggestions mostly editorial. I also read it with eye of a person who is not a native speaker of English. Two points I would like to mention here as well as on attached, PONE-D-20-09801_R1 AUl. Tables and figures has extra informations in titles. Most of them have been shown on attached document. You have the most of information in the text. There is no need repeat them again header fo the table or figure. Please check all of them not only I have mentioned. Some information iunder table can be moved to text, please see the attachment. The first paragraph of discussion is real conclusive paragraph. It should be moved to conclusion.
---

## [Editor Report · Acceptance letter]

16 Jun 2020

PONE-D-20-09801R1 

A herbicide resistance risk assessment for weeds in wheat and barley crops in New Zealand 

Dear Dr. Buddenhagen:

I'm pleased to inform you that your manuscript has been deemed suitable for publication in PLOS ONE. Congratulations! Your manuscript is now with our production department. 

Kind regards, 

on behalf of

Dr. Ahmet Uludag 

Academic Editor

PLOS ONE